# Extraction of Polyphenols from Unripened Coffee (Coffea Arabica) Residues and Use as a Natural Coagulant for Removing Turbidity

Diana Marcela Cuesta-Parra [1,*], Felipe Correa-Mahecha [1], Juan Pablo Rodríguez-Miranda [2], Octavio José Salcedo-Parra [3] and Edwin Rivas-Trujillo [3]

1. Direction of Chemistry and Environment, Faculty of Engineering, University of America Foundation, Bogotá 110311, Colombia; felipe.correa@profesores.uamerica.edu.co
2. Facultad de Medio Ambiente y Recursos Naturales, Universidad Distrital Francisco José de Caldas, Bogota 110711, Colombia; jprodriguezm@udistrital.edu.co
3. Facultad de Ingeniería, Universidad Distrital Francisco José de Caldas, Universidad Nacional de Colombia, Bogota 110231, Colombia; ojsalcedop@unal.edu.co (O.J.S.-P.); erivas@udistrital.edu.co (E.R.-T.)
* Correspondence: diana.cuesta@profesores.uamerica.edu.co

**Abstract:** The coffee agribusiness generates significant amounts of waste that becomes an environmental problem in producing countries. For example, synthetic coagulants have sustainability disadvantages. Immature coffee beans are collected together with mature beans, and their high polyphenol content makes them unsuitable for coffee production and commercialization. This paper aims to test the coagulant activity of polyphenols extracted from *Coffea arabica* residues in synthetic water samples to use them as raw material for producing a natural coagulant based on bioeconomy. It would thus allow immature coffee beans to recover, avoiding their inadequate disposition. An extract was obtained from residual green coffee beans using the ultrasound-assisted separation technique with a mixture of ethanol and water in a 1:1 ratio. The Folin–Ciocalteu method was applied for the total polyphenols quantification, resulting in a concentration of 73.54 ± 0.05 mg GAE (Gallic Acid Equivalent) per gram on a dry coffee basis (GAE/gDB). The synthetic water for the study was prepared with kaolin, showing initial turbidity of 520.90 ± 0.1 NTU (Nephelometric Turbidity Units). First, the effect of pH was determined on the coagulant activity at a fixed dose of polyphenols 2.6 mg GAE/L. Second, the dose and pH results were evaluated using a multilevel factorial design with 5.20, 3.90, 2.60, and 1.30 mg GAE/L doses and pH at 2.5, 3.0, 3.5, and 4.0. Third, the turbidity removal achieved was 99.94% at a dose of 3.9 ± 0.05 mg GAE/L and a pH of 2.5. Fourth, the result was compared with the turbidity removal of the aluminum sulfate dosed at a concentration of 3 mg/L on the same water type, with a pH variation between 5.5 and 8, obtaining 98.69% of turbidity removed. Finally, the research demonstrated that the polyphenols extracted from the residues of the *Coffea arabica* species possess a high electrochemical affinity that would allow removing turbidity by coagulation in waters at specific pH levels with similar removals to those obtained with aluminum sulfate.

**Keywords:** *Coffea arabica*; natural coagulant; ultrasound assisted extraction; turbidity

## 1. Introduction

According to the Ministry of Environment, Housing and Territorial Development in Colombia, 44% of the rural population is supplied with non-potable water, even though Colombian legal regulations require service providers to comply with water quality requirements under physicochemical and biological parameters for consumption. These provisions are not necessarily fully respected [1]. Regarding wastewater, the Superintendency of Public Services estimates that only 42.2% of the flow of water discharged to water

sources in Colombia has had some treatment (Institute of Hydrology, Meteorology and Environmental Studies, 2019 (known as IDEAM in Spanish)).

Turbidity in water is due to suspended particles, which are the prime cause of deterioration in liquid quality as they adsorb different contaminants such as bacteria, nutrients, toxic compounds, and heavy metals. Thus, turbidity is considered one of the most relevant parameters for water quality assessment [2], due to the ease of implementation, low infrastructure requirements, and lower costs. Hence, coagulation and flocculation processes are the most widely used for wastewater and drinking water treatment [3,4]. Coagulation is characterized as a rapid process, which depends on water properties such as pH, temperature, and amount of dissolved and suspended particles, and involves the addition of a chemical or natural substance called coagulant, whose function is to destabilize suspended, colloidal, and dissolved matter [5].

Aluminum sulfate and aluminum polychloride (PAC) are widely used compounds in coagulating wastewater and consumption. Up to 35% of the carbon footprint of a conventional treatment water plant comes from using chemicals such as coagulants; the health and environmental effects of the mining activities for extracting bauxite, the raw material for processing aluminum, have also been reported [6]. On the one hand, the sludge generated during coagulation can reach rates between 4% and 7% concerning the treated water, implying a large amount of material available. It possesses a high aluminum content that may cause secondary pollution in the release during drying processes or the leaching when already landfilled [7] The relationship between the consumption of aluminum and the development of diseases, such as cancer, Alzheimer's, disorders of the bone system, and chromosomal modifications, has been reported [8]. Sludges from ferric coagulant treatments, on the other hand, widely used in sewage coagulation, are often implemented as additives in soil amendments; however, when applied in agricultural systems, they capture phosphorus, reducing availability in the crop and increasing the costs of fertilization, so authors have suggested the search for other methods of disposal [9].

The increasing awareness of the ecotoxicological and human health impacts of synthetic coagulants and flocculants has drawn attention to environmentally friendly approaches to water treatment, available in a growing body of research aimed at developing coagulant products from renewable, biodegradable, and non-toxic resources that boost the circular economy and reduce both the carbon footprint of coagulation and the dependence on chemicals through the partial or total substitution (Saleem & Bachmann, 2019). From a bioeconomy perspective, the production and application of polymeric coagulants of plant or microbiological origin seem to be the most viable alternatives for increasing the sustainability of water purification processes, so research on natural coagulants is a must [2,10].

As natural coagulants, there are the *Moringa oleifera* [11] and *Phaseolus vulgaris*, or the common bean proteins, the different extracts obtained from actinobacteria, starches and their derivatives, the extract tree seed known as chestnut (*Aesculus hyppocastanum*) oak or various species of the genus *Opuntia*, the plant known as nopal (Oladoja, 2015). Polysaccharides obtained from the mucilage of *Plantago major* L. have proved utility in bleaching water contaminated with dyes [12] to the same as that of other species of the genus *Plantago* and polysaccharides of *Enteromorpha* sp. (Chorophyta, macroalgae) [13]; similarly, chitosan obtained from chitin, one of the most abundant polymers on earth, is extracted from various sources such as animals, insects, and fungi; it has an activity for removing turbidity and organic matter of the water [14].

Reports also described that the polyphenols present in some plants have activity in removing dyes in synthetic waters. The ethanol or grape seed extract (*Vitis vinifera*) was evaluated as a cationic coagulant on synthetic waters with malachite green and crystal violet dyes achieving removals greater than 80% with doses of 1.5 g of extract per liter. The samples' gradual discoloration and the appearance of precipitated flocs indicate that these extracts can lead to coagulation and flocculation processes. Thus, catechin and tannic acid have been identified as the compounds responsible for removing the color [15]. Some coag-

ulants derived from polyphenols, from plants such as *Jatropha curcas* (Abidin et al., 2013), *Acacia mearnsii* [16], and *Opuntia* spp. [17], have been tested as alternatives for turbidity removal, treatment of urban wastewater [18–20] and the removal of color in synthetic water (Lopes et al., 2019). The evaluations included the leachate treatment [21] and wastewater from the textile and dairy industry; polyphenols were also modified by gelation to generate adsorbents that remove cationic contaminants or by cationization processes that allow ampholytic coagulants [4].

Coffee, one of the most favored beverages worldwide and the leading commodity on the planet after oil, has polyphenols in each part of the fruit [22], which are related to the aroma, flavor, and color of the drink [23]; so, its residues can be used as a source of antioxidants, nutraceuticals, and preservatives of some food preparations in areas of cosmetology and water treatment [24]. Colombia is Latin America's second-largest producer; only 5% of the weight of the fresh fruit is used for the preparation of the beverage, while the remaining 95% becomes residues in the production chain [25], turning this waste into a potential source of materials for obtaining different products.

The extraction and determination of polyphenol concentrations in coffee residues are conducted by various methods. Patay and her team (2016) evaluated mature, immature beans and their pericarps of *Coffea arabica* using a 40:60 solution of isopropanol-water, then analyzed their polyphenol content by the Folin–Ciocalteu method, finding results of 2.12%, 4.14%, 1.68%, and 1.63% respectively. Prifitis and his colleagues (2018) performed ultrasound extraction of previously ground beans, although they did not mention the species used or the maturity stage: the coffee contained 42.61 mg GAE/g of dry material; the same study identified 3-chlorogenic acid (16. 61 mg/g), 4-and 5-chlorogenic acids (13.62 mg/g). The isolation of bioactive compounds present in the solid matrices of coffee by-products and green coffee has been studied, and the conventional and modern extraction techniques have been applied [26]. Among the traditional techniques is Soxhlet extraction, which has drawbacks due to the amount of solvent required, long extraction times, and in some cases, the loss of the compound to be separated [27]. New techniques include extraction with ionic solids, which uses supercritical fluids by pulsed electric fields assisted with microwaves and ultrasound, methods that improve the yield and obtain purer products [28].

## 2. Materials and Methods

### 2.1. Raw Material and Chemical Substances

Unripe green fruit of the *Coffea arabica* species, provided by the Technological Park Corporation for Coffee and Coffee Growing (Technicafé), located in the municipality of Cajibío, department of Cauca, Colombia, collected during February and March 2019 was used. The sample was kept refrigerated (7 °C) for less than a week to avoid decomposition while extraction of the polyphenols was carried out with ethanol; the quantification of phenols used the Folin–Ciocalteu reagent, sodium carbonate, and gallic acid in the calibration of the standard curve. For the jar test, the pH of the water was modified with sodium hydroxide and hydrochloric acid.

The synthetic water used for the initial turbidity control resulted from an impalpable kaolin solution. The comparison of the natural coagulant to a commercial coagulant used aluminum sulfate.

### 2.2. Ultrasound Assisted Extraction

The polyphenols were extracted from one sample of coffee ground following the methodology described by Cuesta and Correa (2018) with some modifications: 16 consecutive extractions were carried out using a mixture of ethanol-water (50:50) as an extraction agent at a ratio of 1:1 (solid:solution). The ground coffee and the solvent were mixed in 250 mL Schott® bottles, sonicated for 30 min in an ultrasonic bath witeg® Germany Wisd Ultrasonic cleaner Set, model WUC-D06H using a frequency of 60 kHz, afterward, the extracts were collected and vacuum filtered using a Büchner funnel; the polyphenol content

of the filtrate was quantified, and this was dried using a *Heildoph® Germany G3 Hei- Vap precision* rotary evaporator at temperatures between 56 °C and 78 °C and pressures between 556 mbar and 175 mbar for solvent recovery [29].

### 2.3. Determination of Total Phenols of the Extract

Polyphenols extracted from the coffee fruit were quantified through the colorimetric Folin–Ciocalteu method described by Chen et al. (2015). The calibration curve was adjusted using standard solutions of gallic acid; the blank determination was prepared with distilled water, and absorbances were measured by a Jenway® *United Kingdom 6320D* at a wavelength of 765 nm. The concentration of total polyphenols is expressed in milligrams equivalent to gallic acid per liter and gram dry basis of the coffee sample (mg GAE/L and mg GAE/gDB coffee).

### 2.4. Sample Water

The synthetic water for the study was prepared following the methodology described by Miller and her team (2008). Indeed, 5000 mg/L of kaolin were mixed for each liter of water, and the suspension was stable for one hour, achieving turbidity of 520.90 ± 0.1 NTU and an initial pH of 7.32 ± 0.1.

### 2.5. Coagulation

The adaptation and coagulation procedure for the synthetic water was performed under the ASTM D 2035-90 descriptions from Liang and her team (2019) using a conventional digital flocculator. The 1000 mL samples were deposited in beakers, homogenizing them at an instantaneous mixing speed of 120 rpm. The initial turbidity and initial pH were measured, adjusting the working pH with hydrochloric acid or sodium hydroxide according to the different levels of the experimental design. Subsequently, the coagulant dose was added and homogenized for 1 min at 120 rpm. Later, the speed was reduced to 30 rpm, stirring for 20 min, after which the solutions were sedimented. Samples settled in the central part of the vessel to determine the turbidity and the final pH; as a result, the treatment was performed for each studied pH figure subjected to the same conditions but without adding a coagulant.

Turbidity was determined through the Turbiquant® Germany 1100 IR turbidimeter, following the protocol of the standard methods for the analysis of drinking water and wastewater [30]. The performance of the coagulation process was calculated, taking into account the coagulant activity as a percentage of turbidity removal and according to the following equation [3,31].

$$Coagulant\ activity\ (\%) = \frac{T_A - T_R}{T_A} \times 100\%$$

where:

$T_R$ is the residual turbidity after treatment in Nephelometric Turbidity Units.

$T_A$ is the turbidity of the control sample in Nephelometric Turbidity Units; it was handled under the same conditions as the test samples but without adding any coagulant.

A first test established the effect of pH on the percentage removal of turbidity at a fixed dose of 2.60 mg GAE/L, at pH of 2.0, 3.0, 4.0, 5.0, 6.0, 7.0, 8.0, 9.0, 10.0, 11.0, and 12.0, to determine the range of best coagulation performance of the extract.

Once the pH range with the best coagulant performance was known, a factorial experimental design was executed with four pH levels (2.50, 3.00, 3.50, and 4.00), and four dosage levels (5.20, 3.90, 2.60, and 1.30 mg GAE/L), with three repetitions for a total of 48 experimental tests.

Various assays removing turbidity via aluminum sulfate on synthetic water at a dosage of 3 mg/L and pH between 5.5 and 8.0 were conducted, in which improved performance of aluminum sulfate was expected based on previous studies [32,33].

### 2.6. Statistical Analysis

The data obtained from the turbidity removal were processed through an analysis of variance (Anova), using the data processing program SPSS Statistics 24 (IBM), under a 95% confidence level.

## 3. Results and Discussion

### 3.1. Extraction of Polyphenols

Polyphenols from immature coffee of the *Coffea arabica* species were separated using ultrasound-assisted extraction. The concentration of total phenols was 73.54 ± 0.05 mg GAE/gDB coffee, resulting in lower rates than those found in previous research projects with immature coffee beans. Cuesta and Correa (2018) obtained 129.2 ± 0.05 mg GAE/gDB coffee via extraction assisted with ultrasound at 60 kHz for 30 min; in this study, extraction with acetone was performed; however, the difference in extraction cannot be attributed only to the type of solvent since the same samples were not used in the present study. Therefore, the plant samples can have natural variations in the content of secondary metabolites caused by variability in the weather and culture (Hećimović et al., 2011). The use of alcohol mixtures as extraction agents requires applying the principles of green chemistry, which, together with ultrasound-assisted extraction and biorefinery approaches, seek to improve the environmental performance of products for the industry (Cvjetko Bubalo et al., 2018).

### 3.2. Effect of pH on the Removal of Turbidity

Figure 1 describes the results of the tests for removing turbidity at different pH levels and fixed doses of coagulant, 2.60 mg GAE/L to extract coffee residues. For the extract, the best removal turbidities were 95.69 ± 1.00% and 93.56 ± 1.58%, obtaining an acid pH of 2 and 3, respectively. However, between 4 and 10, a significant reduction in the coagulant activity was observed, keeping it below 40%, then there was an increase of 70.23 ± 0.70% at pH 11 and a drop to the minimum at pH 12.

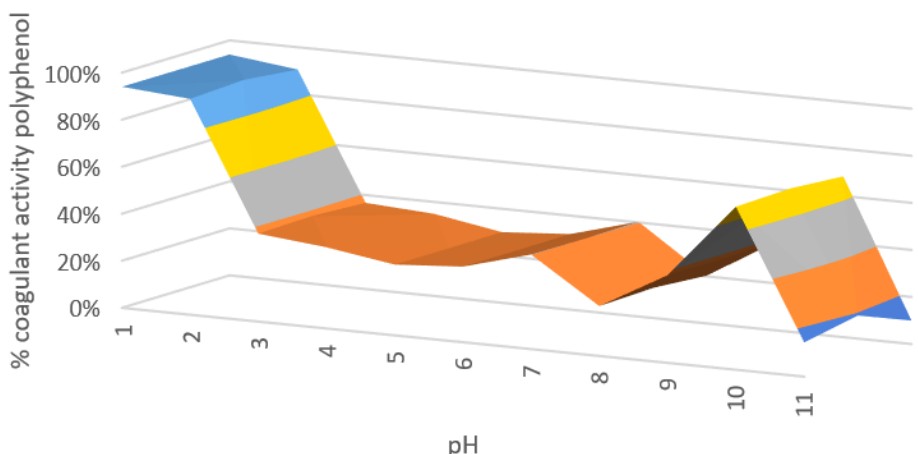

**Figure 1.** The pH effect on the removal of turbidity of the polyphenol extract, 2.6 mg GAE/L dose. Average levels of three determinations, n = 3. The error bars represent ± the standard deviation of the determinations.

Removal tests were conducted as per the design by taking pH, in 10 levels for understanding the removal behavior polyphenol composite. ANOVA (analysis of variance) for turbidity removal is provided in the Table 1. Each pH effect and interaction was studied through analysis of variance; the table shows the variance of factors. The F value was determined by relationship between each pH parameter and removal ratio. F value obtained as 250.8 shows the significance analysis for each pH variation separately.

**Table 1.** ANOVA for relationship between pH and % turbidity removal.

| Source | Sum of Squares | Degrees of Freedom | Mean Square | F-Value | *p*-Value |
|---|---|---|---|---|---|
| Relationship between pH and turbidity removal | 0.00101906 | 1 | 0.00101906 | 2.42830421 | 0.15359017 |
| Intra pH and turbidity removal ratio | 0.94753023 | 9 | 0.10528114 | 250.873741 | $1.0251 \times 10^{-09}$ |
| Pure error | 0.00377692 | 9 | 0.00041966 | | |
| Cor total | 0.95232621 | 19 | | | |

Coagulant activity is influenced by pH because the active agents in coagulants generally carry anionic or cationic charges [5,31]. Korolev and Nesterov (2018) stated that the hydroxylation surface charge regulates the electrochemical behavior of kaolinite and is the most sensitive characteristic of surface charge in clays. They originate from protonation/deprotonation reactions at the edges of the particles, neutralization reactions at the surfaces, consequence of the breakage of the Si-O-Si or Al-o-Al bonds, as well as from the hydrolysis of the oxygen atoms. In their study, they found that at pH less than 3, the clay has a net positive charge; at a pH of around 3, there is a charge change; and at a pH above 3, it has a net negative charge [34,35].

Jeon et al. (2009) hypothesize that the hydroxyphenyl groups of polyphenols can interact with positively and even negatively charged contaminants through dipolar forces resulting from the electronegativity of oxygen and the loss of hydrogen ions in aqueous solutions. Thus, phenolic groups in tannins and other polyphenols cause the anionic character of the molecules (Lopes et al., 2019), whereas in polyphenols, hydroxyl groups possess a net acidic action and generate a higher level of acidity, as occurs with caffeic acid, a relatively abundant compound in coffee. Thus, the deprotonation of these groups produces stable anions due to the formation of bridge patterns and hydrogen [36], which would explain why a higher coagulating activity is achieved at low pH levels, when kaolin and polyphenols have different charges. Complexation and net charge neutralization by affinity between the pollutant and the coagulant are the main mechanisms for the precipitation of solubilized or suspended pollutants in water in the case of polyphenols (Jeon et al., 2009); however, it should be noted that acidification of the solution favors a precipitation mechanism of the clay particles (Aljerf, 2018).

In the case of *Opuntia ficus indica*, it was shown that the phenol group is responsible for the improved coagulation and flocculation activity, finding out that the optimal pH in synthetic waters with kaolin and humic acids was 10; although, there was also activity at acidic pH rates (Bouaouine et al., 2018). At pH 10, clays exhibit a high capacity for anion and cation exchange, so the cationic sites may be present on the surface and allow absorption of anions in conditions of varied pH. This phenomenon occurs due to the interactions of iron and aluminum oxides in kaolin. In another study, Bouaouine and his team (2019) used oil mill wastewaters defatted (a byproduct of olive) as bioflocculant in treating synthetic water prepared with kaolin and humic acids, concluding that the optimal removal with a dose of 100 mg/L was 92% ± 2% at a pH of 11, pH in which the present study found the activity of 70.23% ± 0.7%. The authors explained that tannins and cellulose are active functional groups in charge of flocculation. Similar findings were observed for extracts of *Opuntia ficus indica*, in which an adsorption bridge model was proposed to ionize tannins at very alkaline pH levels, establishing ionic and hydrogen bonds with the kaolin aluminum and the starches so that a contribution to the floc quality through bridges and dragging was achieved [37].

The range of initial pH at which the best performances of the aluminum sulfate were achieved corresponds to those reported by other authors (Zhao et al., 2009). Guo and his colleagues (2015) also found that the coagulant activity of sulfate aluminum had reduced when the pH assumed alkaline figures, as it happens in the present work. For acidic rates, aluminum has positive charges, so the principal coagulation mechanism occurs by charge neutralization.

A second jar test was performed to evaluate the effect of the dose, varying the concentrations of the polyphenolic extract: doses of 1.3 mg/L ± 0.05, 2.6 ± 0.05 mg/L, 3.9 ± 0.05 mg/L, 5.2 ± 0.05 mg/L were applied at pH between 2.5 and 4, similar to the best performances of the previous test. Figure 2 illustrates the results of the four coagulant doses at four different pH; the trend of the removal percentage is shown taking into account the pH used versus the concentration of added polyphenols (mg GAE/L); for pH 2.5 and 3.0, effectiveness greater than 99% is observed for all after added concentrations. The decrease in efficiency from pH 3.5 to almost 90% of removal for 3.9 mg GAE/L is evident. The pH at 4.0 confirms minimal efficiency reaching a removal percentage of less than 50% for the concentration of 2.6 mg GAE/L. These results allow concluding that the predominant efficiency of solid removal in water is visible at pH 2.5 and 3.0 for all concentrations of studied polyphenols. In Figure 2, the error bars correspond to the standard deviations of the means of three replicates per combination between polyphenol doses and pH, indicating that the 3.9 mg GAE/L and the pH 3.5 doses show better variability (standard deviation of 6.09%), followed by 2.6 mg GAE/L and pH 4.0 (3.16%), and 1.3 mg GAE/L and pH 4.0 (2.18%). The other 13 combinations present standard deviations that do not exceed 1.0%, demonstrating that the developed method provides duplicable outcomes.

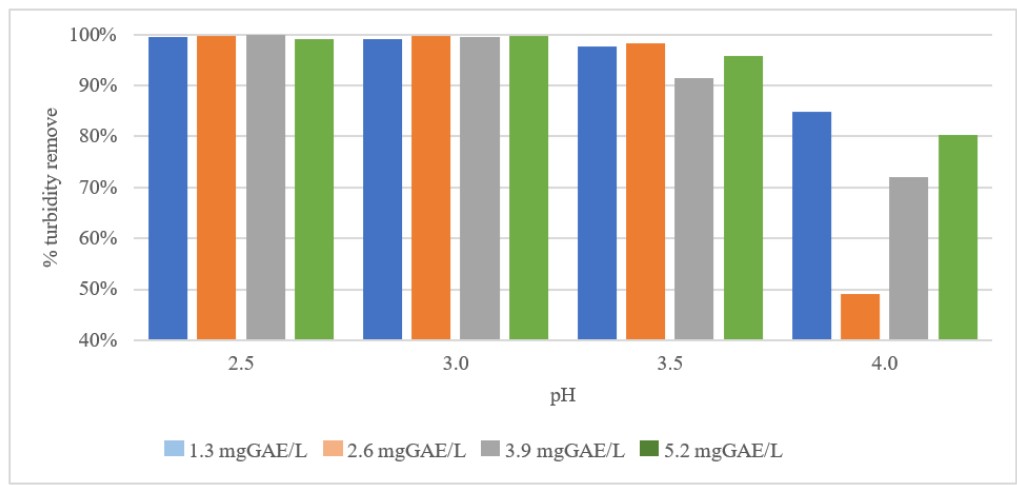

**Figure 2.** Relationship between coagulant dose, pH, and turbidity removal. Figure obtained with the mean removal scores of three replicates for each pH level and dose, n = 3. The error bars represent ± the standard deviation for the determinations.

On the one hand, the results specify that acidic pHs favor flocculation and coagulation, even when we used the lowest concentration of polyphenols. On the other hand, we found that the condition for producing the highest removal percentage corresponds to the combination of 3.9 mg GAE/L with pH 2.5. It is an indication that there is no dose-dependent behavior and that the most determining influence includes the pH of the sample rather than the added concentration.

Considering a *p* score = 0.05 with three replicates for each determination, the analysis of variance concerning the concentration of polyphenols used for the jar test trials (5.2, 3.9, 2.6, and 1.3 mg GAE/L) yielded statistically significant differences ($p < 0.0001$). At pH 4.0, a minimum rate of 45.98% for 2.6 mg GAE/L and a maximum of 87.25% for 1.3 mg GAE/L were obtained (Table 2). The results for the pH ranges studied (2.5, 3.0, 3.5, 4.0) are similar ($p < 0.00015$) to the concentration, indicating that there are differences between the four pHs studied; however, it should be noted that for pHs between 2.5 and 3.5, the percentage of removal remained above 95%, in contrast to the 80% obtained at pH 4.0.

**Table 2.** ANOVA for relationship between acid pH and % turbidity removal.

| Source | Sum of Squares | Degrees of Freedom | Mean Square | F-Value | *p*-Value |
|---|---|---|---|---|---|
| Sample | 0.05208757 | 3 | 0.01736252 | 53.038351 | $1.6178 \times 10^{-12}$ |
| Polyphenol doses | 0.65558436 | 3 | 0.21852812 | 667.551032 | $6.4239 \times 10^{-29}$ |
| Relationship between pH and turbidity removal | 0.18626541 | 9 | 0.02069616 | 63.2217978 | $77438 \times 10^{-18}$ |
| Intra pH and turbidity removal ratio | 0.01047545 | 32 | 0.00032736 | | |
| Cor total | 0.9044128 | 47 | | | |

The analysis of variance for the interactions between the concentration and pH levels used for the research shows statistically significant differences ($p < 0.05$) with a maximum of 99.94% removal, pH 2.5, and 3.9 mg GAE/L. It should be noted that the results found between pH 2.5 and 3.0 were greater than 99% of removal for the four concentrations of used polyphenols (5.2, 3.9, 2.6, 1.3 mg GAE/L). These findings indicate that the polyphenols obtained from the coffee samples show improved turbidity removal at pH below 3.5, which is explained by the difference in charges between the pollutant, kaolin, and polyphenols [37].

In the case of other coagulants with polyphenolic active component, no turbidity removal studies were found at pH levels lower than 3. Bouaouine and his team (2018) evaluated the turbidity removal of *Opuntia ficus-indica* extracts using synthetic water with kaolin and humic acids. They found removals of 92% at pH of 10 to 12; the researchers also reported removals at growing pH levels between 5 and 3, reaching 88% to pH 3. The study suggests that lignin and tannin phenolic groups are responsible for the coagulant action. A later paper reported optimal level at pH 10, dose of 20 mg/L. and coagulant activity of 93% from fractionating the extract. It was concluded that the molecules responsible for the coagulant action correspond to quercetin, a flavonoid that belongs to the polyphenols, and starch. These molecules have a synergistic action in the coagulation–flocculation process at alkaline pH [37]. In the present work, the action of other components that may exert some synergistic action during the coagulation–flocculation process is not ruled out.

Vuppala et al. (2019) report removal of 99.2% and pH of 4.5 using 100 mg/L in wastewaters of the olive oil industry (olive oil mills). It is notable that they did not perform assays to pH below 4.5. Bouaouine and his team (2019) also addressed the reuse of mill defatted waters as bioflocculant in synthetic water containing kaolin. They succeeded in a removal of $92 \pm 2\%$ at pH 11 using a dose of 100 mg/L. So, tannins and starches were identified as the molecules responsible for coagulant activity; the effect was not studied at pH below 3. In this study, a coagulant activity of $70.23 \pm 0.7\%$ was reached at pH 11, indicating the need to explore in the future the removal of contaminants at alkaline pH with extracts obtained from coffee residues

Dela Justina and his team (2018) evaluated a coagulant extracted from the species *Acacia mearnssi*; subsequently cationized in wastewaters from the milk industry in a pH range between 4 and 10. The dosage used was 200 mg/L in the conducted tests, and removals greater than 90% were obtained except at pH 4, finding a relationship between pH and turbidity removal. The cationization of tannins leads to a change in the range of action of polyphenols against contaminants; thus, it is a technique to attain coagulants with better pH range [16]. Sánchez et al. (2010) evaluated the effect of pH on turbidity removal using a coagulant based on modified tannins in surface waters. They found that the coagulation process is more effective at a slightly acidic pH close to 4, removing the turbidity at 95%, which can be explained by the appearance of species with a positive charge or not soluble in water.

Hussain and his team (2019) tested the dust of the pine cones of the *Pinus gerardiana.* The coagulating activity in water with kaolin at different pH levels between 2 and 12 yielded that the best removals were 82% at pH 2 and 76% at pH 12 using doses of 0.2 mL/L of extract; however, the molecular species responsible for the coagulation effect were not

identified, but the cones of this species are known for having an appreciable content of tannins and polyphenols [38].

The coagulant activity of the coffee extracts presented in this work is within the levels reported for other natural coagulants in synthetic waters with kaolin. The *Moringa oleifera* reached a removal of 98% for water with high turbidity (500 NTU) using doses of 400 mg/L of extract [39], while for *Planago ovata* removals of 95.6% are achieved at pH levels below 8 from waters with a turbidity of 250 NTU [40]. It confirms that the extracts of this agro-industrial residue would potentially remove turbidity in clay waters.

## 4. Conclusions

The objective of this research was to evaluate the coagulant activity of extracts of immature *Coffea arabica* residues. So, an ultrasound-assisted extraction using an ethanol-water mixture as solvent was carried out, finding that the contents of polyphenols in the extract reached 73.54 ± 0.05 mg GAE/gDB coffee, with lower figures than those reported in previous studies. The type of used solvent and the agroclimatic variables may influence the content of polyphenols and their extraction.

The pH has a significant influence on the coagulant activity of the extract with the best yields below 3 and with a local maximum pH level of 11. The increase in coagulant activity at acidic pH can be explained by the difference between the charges of kaolin (positive charges at acidic pH) and those of polyphenols (negative charges). Therefore, the coagulation mechanism can occur by charge neutralization, while at pH 11, coagulation is achieved due to the formation of ionic bonds between polyphenols and aluminum oxides on the kaolin surface.

In summary, the highest coagulant activity was 99.88% at a dose of 3.9 mg/L at pH 2.5. From statistical analysis, significant differences between pH levels and doses were found. In all treatments with pH below 3, removals over 98% were reached with no evidence of a dose-dependent effect. Finally, the comparison with the coagulant activity of aluminum sulfate showed a more notorious activity in the coffee residues, although at a different pH level.

**Author Contributions:** Experimentation, methodology D.M.C.-P. and F.C.-M.; conceptualization, J.P.R.-M., O.J.S.-P. and E.R.-T. All authors have read and agreed to the published version of the manuscript.

**Funding:** This research received no external funding.

**Institutional Review Board Statement:** Not applicable.

**Informed Consent Statement:** Not applicable.

**Data Availability Statement:** Data supporting reported results can be found in https://repository.uamerica.edu.co/handle/20.500.11839/6704?locale=es (accessed on 24 January 2022).

**Conflicts of Interest:** The authors declare no conflict of interest.

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
