# Peer review of "Extraction of Polyphenols from Unripened Coffee (Coffea Arabica) Residues and Use as a Natural Coagulant for Removing Turbidity"

_processes, doi:10.3390/pr10061105_

Round 1

Reviewer 1 Report

This paper experimentally investigated the natural polyphenols extracted from unripened coffee residues and the turbidity removal capacity of the polyphenols are also evaluated. They found that the turbidity removal reaches 99.94% at a dose of 3.9±0.05 mg GAE/l and pH=2.5. In addition, the comparison of the turbidity remotion capacity between the natural polyphenols and aluminum sulfate were also carried out. This paper promotes the application of the coffee residues and reveals the turbidity removal capacity of the extracted polyphenols. Some comments are listed as follows:

  1. In this work, the commercial aluminum sulfate is only applied at pH=5~8. Therefore, further examinations should be carried out for pH less than 5 to prove the priority the turbidity removal capacity of the extracted polyphenols.
  2. Apart from the aluminum sulfate, other commercial coagulants should also be tested at the same condition, especially at pH<4.
  3. A brief economic evaluation about the commercial application of the natural polyphenols extracted from unripened coffee residues should be carried out.

Author Response

Bogotá D.C., may 13th 2022

Dears reviewer

Thank you for your comments on our scientific article. We will respond to your corrections below:

Review 1

This paper experimentally investigated the natural polyphenols extracted from unripened coffee residues and the turbidity removal capacity of the polyphenols are also evaluated. They found that the turbidity removal reaches 99.94% at a dose of 3.9±0.05 mg GAE/l and pH=2.5. In addition, the comparison of the turbidity remotion capacity between the natural polyphenols and aluminum sulfate were also carried out. This paper promotes the application of the coffee residues and reveals the turbidity removal capacity of the extracted polyphenols. Some comments are listed as follows:

  1. In this work, the commercial aluminum sulfate is only applied at pH=5~8. Therefore, further examinations should be carried out for pH less than 5 to prove the priority the turbidity removal capacity of the extracted polyphenols.

The Theoretical aluminum solubility diagram  indicates that phase separation occurs at pH 5 to 8, therefore at acidic pH the metals are soluble, therefore in clay content does not precipitate. However, we could not perform the experimentation at acidic pH because of the short time given for corrections, we decided to remove the aluminum sulfate content from the paper.

  1. Apart from the aluminum sulfate, other commercial coagulants should also be tested at the same condition, especially at pH<4.

The theoretical iron solubility diagram shows that insolubility occurs at pH below 4 and pH above 9. Ferric chloride is another commercial coagulant for industrial use that could be evaluated. However, no tests have been carried out with this coagulant.

  1. A brief economic evaluation about the commercial application of the natural polyphenols extracted from unripened coffee residues should be carried out

We are conservative in financial studies because the natural coagulant has not yet been evaluated on industrial production scale. In the future we expect to conduct cost studies

Regards,

The authors

Author Response

Bogotá D.C., may 13th 2022

Dears reviewer

Thank you for your comments on our scientific article. We will respond to your corrections below:

This paper aims to test the coagulant activity of polyphenols extracted from Coffea arabica residues in synthetic water samples to use them as raw material for producing a natural coagulant based on bioeconomy.

Thematically the work is interesting for the researchers and professionals and the proposed manuscript is relevant to the scope of the journal. However, I would expect some mathematical model, at least an ANOVA analysis to test the parameters shown on Fig. 2 or Fig. 1.

Tables 1 and 2, results of the Anova analysis of the experimental process (in yellow) are included in the attached document.

I found it appropriate for publication in the Processes journal, but only after some modifications and clarification from the Authors.

The overall organization and structure of the manuscript are appropriate. The paper is well written and the topic is appropriate for the journal.
The aim of the paper is well described and the discussion was well approached, its results and discussion are correlated to the cited literature data.

The literature review is comprehensive and properly done.

The novelty of the work must be more clearly demonstrated.

The significance of the Work: Given the large number of analyzed data, this is an interesting study with a possible significant impact in this area.

Statistical interpretation of the analytical data must be more properly presented.

Other Specific Comments: The work is properly presented in terms of the language. The work presented here is very interesting and well done, it is presented in a compact manner.
In general, there are no doubtful or controversial arguments in the manuscript. The methodology applied in the research is presented in clear manner, so that it is repeatable by other authors.

The main drawback of the paper is the lack of mathematical model and the discussion regarding the influence of parameters, also the novelty in the present work should be more clearly presented, compared to the works of other researchers?

The novelty of the work is the use of coffee residues; none of the researchers cited above have carried out tests with this species. More than 50 countries in the world produce coffee, but their waste is discarded, while alternatives are being sought, the use of organic waste in the treatment of acidic wastewater.

In my opinion, the authors should put additional effort to demonstrate that the present work gives a substantial contribution in the research area.

Reviewer 3 Report

This paper aims to test the coagulant activity of polyphenols extracted from Coffea arabica residues in synthetic water samples to use them as raw material for producing a natural coagulant based on bioeconomy.

Thematically the work is interesting for the researchers and professionals and the proposed manuscript is relevant to the scope of the journal. However, I would expect some mathematical model, at least an ANOVA analysis to test the parameters shown on Fig. 2 or Fig. 1.

I found it appropriate for publication in the Processes journal, but only after some modifications and clarification from the Authors.

The overall organization and structure of the manuscript are appropriate. The paper is well written and the topic is appropriate for the journal.
The aim of the paper is well described and the discussion was well approached, its results and discussion are correlated to the cited literature data.

The literature review is comprehensive and properly done.

The novelty of the work must be more clearly demonstrated.

The significance of the Work: Given the large number of analyzed data, this is an interesting study with a possible significant impact in this area.

Statistical interpretation of the analytical data must be more properly presented.

Other Specific Comments: The work is properly presented in terms of the language. The work presented here is very interesting and well done, it is presented in a compact manner.
In general, there are no doubtful or controversial arguments in the manuscript. The methodology applied in the research is presented in clear manner, so that it is repeatable by other authors.

The main drawback of the paper is the lack of mathematical model and the discussion regarding the influence of parameters, also the novelty in the present work should be more clearly presented, compared to the works of other researchers?

In my opinion, the authors should put additional effort to demonstrate that the present work gives a substantial contribution in the research area.

Author Response

Bogotá D.C., may 13th 2022

Dears reviewers

Thank you for your comments on our scientific article. We will respond to your corrections below:

The paper contains the corrections requested by reviewer 3

Round 2

Reviewer 2 Report

ITs ok